

# Intercomparison of NOx emission inventories over East Asia

Jieying Ding[1,2], Kazuyuki Miyazaki[3,4], Ronald Johannes van der A[1,5], Bas Mijling[1], Jun-ichi Kurokawa[6], SeogYeon Cho[7], Greet Janssens-Maenhout[8], Qiang Zhang[9], Fei Liu[1], Pieternel Felicitas Levelt[1,2]

[1]Royal Netherlands Meteorological Institute (KNMI), De Bilt, the Netherlands
[2]Delft University of Technology, Delft, the Netherlands
[3]Japan Agency for Marine-Earth Science and Technology, Yokohama 236-0001, Japan
[4]Jet Propulsion Laboratory-California Institute of Technology, Pasadena, USA
[5]Nanjing University of Information Sciences and Technology, Nanjing, China
[6]Asia Center for Air Pollution Research, Niigata, 950-2144, Japan
[7] Department of Environmental Engineering, Inha University, Inchon, South Korea
[8]Institute for Environment and Sustainability, Joint Research Centre, Ispra, Italy
[9]Department of Earth System Science, Tsinghua University, Beijing, 100084, China

*Correspondence to*: Jieying Ding (jieying.ding@knmi.nl)

**Abstract.** We compare 9 emission inventories of nitrogen oxides including four satellite-derived NOx inventories and the following bottom-up inventories for East Asia: REAS (Regional Emission inventory in ASia), MEIC (Multi-resolution Emission Inventory for China), CAPSS (Clean Air Policy Support System) and EDGAR (Emissions Database for Global Atmospheric Research). Two of the satellite-derived inventories are estimated by using the DECSO (Daily Emission derived Constrained by Satellite Observations) algorithm, which is based on an extended Kalman filter applied to observations from OMI or from GOME-2. The other two are derived with the EnKF algorithm, which is based on an ensemble Kalman Filter applied to observations of multiple species using either the chemical transport model CHASER and MIROC-chem. The temporal behaviour and spatial distribution of the inventories are compared on a national and regional scale. A distinction is also made between urban and rural areas. The intercomparison of all inventories shows good agreement in total NOx emissions over Mainland China, especially for trends, with an average bias of about 20% for yearly emissions. All the inventories show the typical emission reduction of 10% during the Chinese New Year and a peak in December. Satellite-derived approaches using OMI show a summer peak due to strong emissions from soil and biomass burning in this season. Biases in NOx emissions and uncertainties in temporal variability increase quickly when the spatial scale decreases. The analyses of the differences show: the importance of using observations from multiple instruments and a high spatial resolution model for the satellite-derived inventories, while for bottom-up inventories, accurate emission factors and activity information are required. The advantage of the satellite derived approach is that the emissions are soon available after observation, while the strength of the bottom-up inventories is that they include detailed information of emissions for each source category.



## 1. Introduction

Emission sources are one of the crucial drivers for a chemical transport model (CTM). Accurate spatial and temporal emission distributions of air pollutants are important for air quality simulations and forecasts (Ma and van Aardenne, 2004; Eder et al., 2009; Zhang et al., 2012; Struzewska et al., 2016).

Up to date emission information is also needed to help policy makers for efficient regulations to control air pollution. In general, two approaches are used to develop emission inventories. One approach is based on statistics combining local information such as emission activity rates and factors of different source categories (Streets et al., 2003). Here we refer to the inventories using this approach as bottom-up inventories. This method results in detailed information on the type, source sector, fuel and

technology. The country-specific emissions are distributed in space with the location of the emissions using representative proxy data. Advances on temporal distribution are still needed to obtain representative maps that go beyond monthly resolution. However, large uncertainties are often introduced due to the uncertainties on all the input parameters in the calculation (Jaegle et al., 2005; Castellanos et al., 2014; Zheng et al., 2014; Li et al., 2016; Saikawa et al., 2016; Li et al., 2017). It is

also time consuming to collect all required information. Another approach to construct emission inventories is inverse modelling using satellite observations to constrain emissions by reducing the discrepancy between the modelled and observed concentrations and taking model and observation errors into account (Müller and Stavrakou, 2005; Konovalov et al., 2006; Sofiev et al., 2009; Miyazaki et al., 2012; Mijling et al., 2013; Streets et al., 2013; Stavrakou et al., 2016). In this study, the

emissions derived with this approach are referred to as satellite-derived emissions. Since observations from satellite instruments like the Global Ozone Monitoring Experiment-2 (GOME-2) and the Ozone Monitoring Instrument (OMI) achieve near global coverage in a single day, we expect that emissions derived from these satellite observations are well constrained on a daily and global basis. A limitation of satellite-derived emission inventories is the difficulty to distinguish emissions from different source

categories.

Nitrogen oxides ($NO_x=NO_2+NO$) play an important role in the formation of tropospheric ozone and secondary nitrate aerosols and in climate change (Jacob et al., 1996; Shindell et al., 2009). The emissions of air pollutants increased rapidly during the last two decades in East Asia due to the rapid economic development. Satellite observations as evidence show a strong increasing trend of $NO_2$

column concentrations since 1995 in China (Irie et al., 2005; Richter et al., 2005; van der A et al., 2006). Akimoto and Narita (1994) built the first regional $NO_x$ emission inventory for Asia on a resolution of 1°×1°. Van Aardenne et al. (1999) estimated $NO_x$ emissions from 1990 to 2020 based on an energy consumption scenario to illustrate the situation of the rapid increase of $NO_x$ emissions in Asia. The early-stage emission inventories including the Asian region had large inaccuracies because

of the insufficient information on emission factors in this region, which were often based on information obtained from studies conducted in European or North American countries (Ma and van Aardenne, 2004). Several follow-up studies have been carried out to derive improved emission inventories for this region. Streets et al. (2003) constructed a comprehensive regional emission inventory with more species in support of the TRACE-P (Transport and Chemical Evolution over the

Pacific) mission for Asia in 2000. However, Wang et al. (2004) concluded that the $NO_x$ emissions of



the TRACE-P inventory are largely underestimated, especially for China, compared to the emissions constrained by measurements from ground stations and aircrafts. Zhang et al. (2009) developed an updated Asian inventory for the INTEX-B (Intercontinental Chemical Transport Experiment-Phase B) mission in 2006, which was based on the TRACE-P inventory but with refined temporal and spatial

resolution. In 2007, a long-term Asian inventory REAS (Regional Emission inventory in ASia) was developed by Ohara et al. (2007) to analyse the trend of emissions. Kurokawa et al. (2013) updated the REAS inventory to a higher spatial and temporal resolution. Lee et al. (2011) built the first version of the South Korean national emission inventory CAPSS (Clean Air Policy Support System) using more detailed local information for emission activities and factors. Tsinghua University developed the

Chinese inventory MEIC (Multi-resolution Emission Inventory for China) based on earlier work of Zhang et al. (2009) (http://www.meicmodel.org). Recently, (Li et al., 2017) constructed the new Asian inventory MIX by combining different regional and national inventories including REAS version 2.1, CAPSS, and MEIC. Zhao et al. (2011) concluded based on Monte Carlo simulations that the uncertainties of bottom-up emissions of $NO_x$ are about -13% to 37% in China. The major uncertainties

are due to oversimplified source classifications and roughly estimated emission factors. The main anthropogenic emissions of $NO_x$ in China are from transport and coal-fired power plants (Liu et al., 2015; Saikawa et al., 2016; Li et al., 2017). Because of the rapid implementation of new technologies and air quality control regulations for power plants and vehicles in China, their emission factors and activities are also changing with time, which makes the bottom-up emission estimates more uncertain

(Zheng et al., 2014; Liu et al., 2015).

Satellite observations of atmospheric species can be used to closely monitor changes in emissions, such as the trend, seasonality, and diurnal cycles of emissions (Streets et al., 2013). For example, with satellites it was observed that $NO_x$ emissions in China started to decrease after 2011 (Gu et al., 2013; de Foy et al., 2016; Krotkov et al., 2016; van der A et al., 2017) as a result of the national regulations

for denitrification equipment at power plants. Mijling and van der A (2012) for the first time derived high resolution (0.25° × 0.25°) emissions over East Asia from satellites using an advanced inverse method called DECSO (Daily Emission estimates Constrained by Satellite Observations) based on an extended Kalman Filter. Ding et al. (2015) demonstrated that this approach is able to detect the monthly change of $NO_x$ emissions due to air quality regulations on a city level. Miyazaki et al. (2012)

derived the first global $NO_x$ emission estimates using an inversion technique based on an ensemble Kalman Filter and improved this method to constrain $NO_x$ emissions by using satellite observations of multiple species (Miyazaki and Eskes, 2013; Miyazaki et al., 2017). This method is referred to as EnKF in this paper.

All emission inventories, both bottom-up and satellite-derived, are facing the same challenge of

validation since it is difficult to directly measure emissions on the ground on such large scales. A common way to validate emission inventories is using them in a chemical transport model to simulate $NO_2$ column concentrations and compare these with in-situ or satellite observations. In this way, however, the validations are highly related to the model performance, which may result in inconsistent conclusions (Zhao et al., 2011).





comprehensive collection of regional bottom-up inventories for East Asia: REAS v2.1 and an interim version of 2.2, MEIC, CAPSS and EDGAR v4.3.1 (Emissions Database for Global Atmospheric Research). The global HTAP v2 is not used because in our domain it is identical to MIX, which

includes REAS, MEIC, and CAPSS. To evaluate the effect of the satellite instrument, we compare two emission data sets from DECSO applied to GOME-2 and OMI observations. To examine the effect of the forecast model performance in the inversion, we compare two emission datasets from EnKF using different CTMs. The intercomparison of 9 emission inventories is presented for the time period 2000-2015 for East Asia on a 0.25° × 0.25° resolution. Figure 1 shows the emission maps in 2008 of all

inventories used in this study, including an average of emissions from all these inventories over the selected domain (102-132°E, 18-50°N) . The description of all emission inventories used in this study will be presented in section 2. Section 3 shows the difference of the spatial and temporal distribution among the 9 inventories. The uncertainties of the inventories are discussed in section 4.

## 2. Emission inventories

### 2.1 Bottom-up inventories

#### 2.1.1 EDGAR

The Emissions Database for Global Atmospheric Research (EDGAR) (Janssens-Maenhout et al., 2013) is a global bottom-up emission inventory using consistent methodology allowing straightforward implementation of scenario assumptions. Emission calculations of the latest version, v4.3.1, are based

on information of international energy balances of IEA (Energy Statistics of OECD and Non-OECD Countries, 2014) and agricultural statistics of FAO and other national or regional statistical information. EDGAR classifies emissions into 12 source sectors: energy, combustion in manufacturing industry, industrial processes and product use, oil production and refining, fossil fuel fires, road transport, non-road ground transport, aviation, shipping, agricultural waste burning, residential and

others. It provides global gridded maps of sector specific and historical emission data from 1970 to 2010 (monthly for 2010) with a high spatial resolution of 0.1° × 0.1°. A detailed description of EDGAR v4.3.1 can be found in Crippa et al. (2016). In this study, we use the yearly emissions of $NO_x$ from 2000 to 2010 integrated to a resolution of 0.25° × 0.25° on our study domain.

#### 2.1.2 REAS

The Regional Emission inventory in ASia (REAS) v2.1 (Kurokawa et al., 2013) provides $NO_x$ emission from 2000 to 2008 at a 0.25° × 0.25° horizontal resolution. REAS v2.1 is based on the previous version REAS v1.1 with updates of activity data and parameters as well as an improvement of temporal and spatial resolution. It enlarges the domain by adding Central Asia and the Asian part of Russia. REAS v1.1 was developed by Ohara et al. (2007) based on the methods described in Streets et al. (2003),

which provided historical and future projected emissions from 1980 to 2020 on a 0.5° × 0.5° resolution for East, Southeast and South Asia . REAS contains calculated $NO_x$ emissions for the source sector of energy, industry, transport, domestic, agricultural activities and soil. We use $NO_x$ emission data of REAS v2.1 and an interim version of 2.2 (hereafter REAS v2.2 for convenience) of 2005 to



2010 to expand the time series of the current REAS inventory. REAS v2.1 includes shipping, aviation emissions that are taken from EDGAR v4.2. In REAS v2.2, soil emissions and all emissions over central and Russian Asia are not included.

### 2.1.3 MEIC

Tsinghua university in Beijing has developed the Multi-resolution Emission Inventory for China (MEIC) model to generate an anthropogenic emission inventory for Mainland China with a spatial resolution of 0.25°. $NO_x$ emissions are presented in four sectors: energy, industry, transport and residential. The China coal-fired Power plant Emission Database (CPED) is used for the power plant sector. CPED includes the latest detailed information of emission factors, activity, locations, etc., and

takes emission regulations into account as well (Liu et al., 2015). In the transportation sector, vehicle population and emission factors at county level are combined with a digital road map, vehicle and road type to derive high resolution on-road transportation emissions (Zheng et al., 2014). More detailed information of MEIC is described in He (2012) and (Li et al., 2017). We use monthly $NO_x$ emissions of MEIC v1.2 from 2007 to 2012.

### 2.1.4 CAPSS

The Clean Air Policy Support System (CAPSS) is a Korean Emissions Inventory System, which provides annual air pollutant emissions with a spatial resolution of 1 km in South Korea as described by Lee et al. (2011). CAPSS derived point source emissions from both real-time air pollutant emission measurements and statistics based on emission factors and activity data. The emission source sectors

are classified into 12 sectors, which are the same as for EDGAR. The latitude and longitude are given for each point source to indicate their location. Area emissions are downscaled by using city-province level activity data including a spatial allocation index database. On-road mobile emissions are calculated by using VKT (vehicles kilometres travelled) and spatially allocated by using traffic volume information for each road. In this study, we use CAPSS $NO_x$ emissions from 2001 to 2013 and re-grid

the data to a resolution of 0.25° × 0.25°.

### 2.1.5 MIX

MIX is a mosaic Asian anthropogenic emission inventory developed by the Model Inter-Comparison Study for Asia (MICS-Asia) and the Task Force on Hemispheric Transport of Air Pollution (TF HTAP) projects (Li et al., 2017). Five emission inventories of different regions are combined in MIX by

normalizing source categories, species, and spatial and temporal resolution of each inventory and provides data in a consistent format. The five inventories are REAS v2.1, MEIC v1.0, PK- $NH_3$ (a high resolution $NH_3$ emission inventory developed by Peking University), ANL-India (an Indian emission inventory developed by Argonne National Laboratory) and CAPSS. The spatial resolution of MIX is 0.25° × 0.25°. We use $NO_x$ emissions from MIX in 2008 to represent both MEIC and CAPSS for the

spatial comparison in this study.



### 2.2 Satellite-derived inventories

#### 2.2.1 DECSO

Daily Emission estimates Constrained by Satellite Observations (DECSO) is an inverse modelling method to update daily emissions of $NO_x$ based on an extend Kalman Filter (Mijling et al., 2013). $NO_x$

emissions are constrained by combining simulated $NO_2$ column concentrations of a regional CTM with satellite observations. The essential part in the inverse calculation is deriving the sensitivity of the $NO_2$ column concentrations on $NO_x$ emissions. A simplified isobaric surface 2-D trajectory analysis is used to take into account the transport of $NO_2$ from the source for a fast sensitivity calculation. In the Kalman filter, the emissions are assumed to follow a persistent model, which expresses that emissions

of tomorrow will be equal to emissions of today. The Eulerian regional off-line CTM CHIMERE v2013 (Menut et al., 2013) is used to obtain simulated $NO_2$ concentrations based on a priori emissions. Note that emissions derived with DECSO become independent from the a prior emissions after a spin-up time of about 3 months. CHIMERE is implemented on a $0.25° \times 0.25°$ horizontal resolution of the region for East Asia (102-120 E, 18-50N) driven by the European Centre for Medium-Range Weather

(ECMWF) operational forecast. The model is set up with 8 vertical layers from the surface to 500 hPa. The CHIMERE simulated columns are extended from 500 hPa to the tropopause with a climatological partial column (2003-2008 average) simulated by the global CTM TM5 for comparison with satellite observed tropospheric columns. The updates of emissions are related to the difference in $NO_2$ between the CTM simulation and satellite observations. A detailed description of DECSO can be found in

Mijling et al. (2013), while the latest improvements are described in Ding et al. (2015, 2017).

DECSO v5 has been applied to observations from OMI (Ozone Monitoring Instrument) and GOME-2 (Global Ozone Monitoring Experiment-2). OMI is a Dutch-Finnish instrument aboard NASA's EOS-Aura satellite (Levelt et al., 2006). The pixel size is $24 \times 13$ km$^2$ at nadir and increases to about $150 \times 28$ km$^2$ at the edge of the swath. The overpass time of OMI at the equator is about 13:30 local time. We

use the tropospheric $NO_2$ column data of the Dutch OMI $NO_2$ retrieval (DOMINO) algorithm version 2 (Boersma et al., 2011). GOME-2 is aboard the sun-synchronous satellite MetOp-A with a local overpass time of around 9:30. The pixel size of the observation was $80 \times 40$ km$^2$ until 15 July 2013. Afterwards, the scan width of the orbit is halved and the pixel size is changed to $40 \times 40$ km$^2$. Tropospheric $NO_2$ columns of GOME-2 are also retrieved with the algorithm DOMINO v2.

Observations are selected with a surface albedo lower than 20% and a cloud radiance fraction lower than 70%. The observations with clouds below 800 hPa are excluded. For OMI data, the pixels affected by the so-called row anomaly (KNMI, 2012) and the four pixels at each side of the swath are filtered out as well. The filter criteria are based on the analyses of Ding et al. (2015, 2017) to reduce the amount of low quality retrievals. The monthly DECSO v5 data set used in this study is based on OMI

satellite observations for the period of 2007 to 2015 (DECSO-OMI) and based on GOME-2 satellite observations for the period of 2008 to 2015 (DECSO-GOME2a). The data is available on www.globemission.eu.



### 2.2.2 Ensemble Kalman Filter (EnKF)

Miyazaki et al. (2012) developed a data assimilation system to estimate global $NO_x$ emissions based on an ensemble Kalman filter technique, which combines satellite observations with a global CTM. In this paper, we refer to this method as EnKF. In this approach, surface $NO_x$ emissions are included in the state vector together with other variables such as lightning $NO_x$ sources and $NO_2$ concentrations. The dependence of $NO_2$ concentrations on $NO_x$ emissions (including complicated chemistry and transport) is taken into account, by using a background error covariance estimated from ensemble CTM forecasts. Miyazaki and Eskes (2013) demonstrated improved $NO_x$ emission estimates by assimilating multiple species (including $O_3$ from TES and MLS, $HNO_3$ from MLS, CO from MOPITT). In this approach, the assimilation of non-$NO_2$ measurements (e.g., $O_3$ and CO) influences largely the concentration and chemical lifetime of $NO_x$ and thus the surface $NO_x$ emission estimates. Miyazaki et al. (2017) updated the EnKF system to combine $NO_2$ retrievals from OMI, GOME2 and SCIAMACHY together with the non-$NO_2$ measurements to optimize the diurnal emission variability. SCIAMACHY (aboard ENVISAT) (Bovensmann et al., 1999) operated from 2002 till 2012 with a local overpass time of 10:00 AM and global coverage every 6 days. The tropospheric $NO_2$ columns are retrieved with DOMINO v2 (Boersma et al., 2004). Observations with a cloud radiance reflectance lower than 50% were used. Two emission datasets are derived using this approach with two different global CTMs: CHASER (Sudo et al., 2002) and MIROC-Chem (Watanabe et al., 2011) (referred to as EnKF-CHASER and EnKF-MIROC respectively) for 2005 to 2015.

CHASER is coupled with an atmospheric general circulation model, CCSR/NIES/FRCGC AGCM v5.7b on a horizontal resolution of 2.8° (T42) and 32 vertical levels on the sigma vertical coordinate system from the surface to 4 hPa (Miyazaki and Eskes, 2013). The AGCM fields were nudged toward NCEP/DOE-II reanalysis (Kanamitsu et al., 2002). The MIROC-Chem model (Watanabe et al., 2011) was developed based on CHASER, with many updates on the tropospheric chemistry and including stratospheric chemistry. MIROC-Chem is coupled to the atmospheric general circulation model MIROC-AGCM v4, on 2.8° (T42) horizontal resolution and it uses the hybrid terrain-following pressure vertical coordinate system with 32 vertical levels from the surface to 4.4 hPa (Miyazaki et al., 2017). The MIROC-AGCM fields were nudged towards the 6 hourly ERA-Interim (Dee et al., 2011) reanalysis.

In both calculations, the a priori $NO_x$ emissions are obtained from EDGAR v4.2 for anthropogenic emissions, GFED v3.2 (van der Werf et al., 2010) for biomass burning emissions and GEIA (Yienger and Levy, 1995) for soil emissions. For the comparison, the monthly emissions on 2.8° resolution from EnKF-CHASER and EnKF-MIROC are linearly interpolated to a horizontal resolution of 0.25° × 0.25° based on emission distributions of the a priori inventory MIX. Note that the shipping emissions near the coast areas are interpolated to land since they are not included in MIX. This could lead to overestimation of emissions over land near busy shipping lanes.



### 3. Intercomparison of NO$_x$ emissions

### 3.1 Temporal evaluation

### 3.1.1 Time series analysis

Figure 2 shows the comparison of all emission inventories as a function of time over Mainland China
(in this paper limited to the study domain shown in Figure 1) for the period 2000-2015. The time series
of the annual total emissions show a large variation among the inventories. The bottom-up inventories
use similar approaches, but the statistical data and assumptions on the penetration and specification of
the present technologies used for their calculations are different and this explains the diversity among
the bottom-up inventories. The differences among the satellite-derived emission inventories are caused
by different inversion techniques, satellite observations and CTMs. The emissions derived by DECSO
with OMI observations show a large discrepancy with the other inventories. Over Mainland China (Fig.
2), all inventories follow a similar trend but with large biases. The NO$_x$ emissions dramatically increase
since 2000 and start to decrease around 2011. All satellite-derived inventories show a decrease in 2015
of at least 20% compared to 2014. The average annual total emissions of 2008 (the only year included
in all inventories) are about $5 \pm 1$ Tg N yr$^{-1}$ for Mainland China. REAS v2.1 presents the highest value
of 7.3 Tg N yr$^{-1}$, while DECSO-OMI shows the lowest value of 4.3 Tg N yr$^{-1}$. The standard deviation
of the emissions is about 20%.

Figure 3 shows the comparison of the temporal variation relative to the emissions in 2008. REAS v2.1,
v2.2 and EDGAR v4.3.1 show a continuous increase from 2000 to 2010 with an annual increase rate of
about 8% on average. NO$_x$ emissions of China in 2010 are almost doubled compared to emissions in
2000. The change rates derived with two EnKF emissions are lower than for REAS and EDGAR
between 2005 and 2010. MEIC and satellite-derived emissions show a slow change of NO$_x$ emissions
from 2007 to 2009 and a large increase in 2010 until 2011, which is consistent with satellite
observations (Gu et al., 2013; de Foy et al., 2016; Krotkov et al., 2016), whereas REAS and EDGAR
reveal a continuous increase during 2007 to 2010. The increase rate in 2011 of DECSO-OMI is about
10% higher than of the other inventories. From 2012-2015, the satellite-derived emissions show a
decrease of total NO$_x$ emissions over China by about 30%. Nevertheless, the peak year is different
between the satellite-based approaches. The DECSO inventories show a decrease starting in 2012,
while the EnKF inventories reach their peak in 2011.

Over a small domain, the discrepancies among the inventories is larger. As an example, Figure 4 shows
the comparison of the emission time series from eight inventories in South Korea. We see differences
by a factor of up to 2 between the inventories in 2008 (the highest one is REAS v2.2 and the lowest one
is CAPSS). The bottom-up inventories EDGAR and REAS show higher emissions than the satellite-
derived emissions. The satellite-derived inventories are generally in closer agreement to the South-
Korean CAPSS than to the other bottom-up inventories. These results highlight the large differences in
the emission trends over a small region like South Korea, even derived with a similar approach.

The satellite-based approaches can be used to make near-real time datasets and to extend the emission
record over a longer period until 2015. Figure 5 (left) displays the peak year of the averaged NO$_x$


emissions from the four satellite-derived inventories. Beijing and Shanghai reach their peak years as early as 2010 and 2008 respectively. Most of the developed provinces reach maximum emissions in 2011 or 2012. In some less developed provinces, emissions were still growing until 2014. To analyse the correlation of the time series of the four satellite-derived inventories for each province, we calculate

the temporal correlation coefficient (R) from each combination of the four emission datasets. Figure 5 (right) shows the minimum correlation coefficient of all combinations. The satellite-derived emission inventories are in good agreement for most of the provinces, with a correlation coefficient higher than 0.6. The provinces with smaller areas often have lower R-values. This means that the uncertainties of $NO_x$ emissions can be notably high on a small area or provincial level.

**3.1.2 Monthly variability**

We calculate the mean monthly emissions of each inventory for its own available time period and normalize them to get the general monthly variability. Figure 6 shows the monthly variability of total $NO_x$ emissions for 7 inventories over Mainland China. MEIC, REAS v2.1 and DECSO-GOME2a show a relatively weak variability while the other four inventories show a distinct summer peak. The EnKF

inventories show a sharp peak in June. REAS v2.1 and DECSO-OMI reveal a peak in July. The summertime peak is probably introduced by enhanced biogenic (e.g., from soils) emissions. The summertime enhancement is considered to be better detected by the OMI afternoon measurements than the GOME2 morning measurements (Boersma et al., 2009), because they are generally maximized in the afternoon. The different representation of the summertime peak between the satellite-derived

inventories will be discussed in Section 4.4. It is notable that all inventories show that $NO_x$ emissions are about 10% lower in February than in the surrounding months, which is due to the Chinese New Year (Ding et al., 2015). All inventories also show a peak in December when heating and electricity consumption is usually high in China.

To study the temporal correlation of monthly different inventories, we use monthly emissions over

Mainland China of the common period (2008 to 2010, i.e. 36 months) for DECSO, EnKF, MEIC, and REAS v2.2. Table 1 shows the correlation coefficient of monthly total emissions from different inventories over Mainland China from 2008 to 2010. The two DECSO inventories show a relative good correlation with MEIC and REAS. The correlation coefficient between the two DECSO inventories is only 0.75 as a result of the different seasonality (see Figure 6). This implies that the observed

seasonality can be highly affected by the choice of satellite observations, reflecting differences in the local overpass time and pixel size. The two EnKF inventories based on the same observations but using different forecast models show a strong correlation. However, for most cases on a provincial or a small regional level, no significant correlation of monthly variability from different approaches can be found (not shown). Due to the coarse resolution of the EnKF emission estimates (i.e. 2.8°), the seasonality

can be strongly mixed up between urban and rural areas within a coarse model grid. The uncertainty in temporal variability of $NO_x$ emissions becomes larger with decreasing spatial scale of emissions, especially where surface types are inhomogeneous.





### 3.2 Spatial distribution

In this section, we choose the year 2008 to compare the bias and spatial distribution, since it is the only year that is contained by all inventories. Because DECSO inventories have a grid that is half a grid cell shifted compared to the other inventories, the comparison per grid cell could theoretically result in up to 40% discrepancy in correlation coefficients due to re-gridding. The calculated discrepancy decreases to less than 10% on the resolution of one degree or more. For a fair comparison, we check the correspondence in spatial distribution on a provincial level. We calculate the correlation coefficient (R) of each two inventories on a regional or provincial level (in total 33 regions including 29 Chinese provinces, South Korea, North Korea, and parts of Vietnam (43%) and Mongolia (53%), as shown by Figure 1. Note that our domain covers only a part of the provinces Heilongjiang (74%), Inner Mongolia (83%), Sichuan (51%) and Yunnan (36%). The comparison result is summarized in Table 2. We use the MIX inventory for 2008 to represent both MEIC and CAPSS depending on the region of interest, since MIX covers the whole domain. All correlation coefficients are above 0.8, which means that the spatial distribution on a regional or provincial level have a good agreement among all inventories. The emission correlations are higher when the emissions are derived with similar methods. Both DECSO inventories have the best correlation with the EnKF inventories, with R-values around 0.95. EnKF-CHASER and EnKF-MIROC show a similar spatial distribution. For the two EnKF inventories, the original 2.8-degree resolution data was subsequently re-gridded to a finer resolution using the same fine-scale distribution. Both REAS inventories have very high correlations (R values are 0.98 and 0.99) with MIX since the same provincial activity data of China are used for calculating emissions in MIX and in REAS v2.1 (Kurokawa et al., 2013). The correlation between EDGAR v4.3.1 and other bottom-up emission inventories is relatively low (R<0.85). EDGAR v4.3.1 uses the emission information from IEA (2014) statistics (before the Coal statistics revision of China) and CARMA v3.1 for distribution while MIX and REAS are based on national statistics, which may lead to the discrepancy.

Both the bottom-up and satellite-derived inventories show large biases but their spatial distributions are similar on a provincial scale. To further examine the spatial distribution and biases, we combine the emissions in bins of one-degree latitude and compare the longitudinal total emissions over land in the study domain. Figure 7 shows that all inventories have similar patterns over latitude but with large biases. The standard deviation averaged over the latitudes (20N-50N) between all the inventories is 27%. For areas, north of 40 N°, the satellite-derived emission inventories are lower than the bottom-up inventories. The relative standard deviation is also large (from 40% to 80%) at high latitudes.

Note that the satellite-derived emissions are total surface emissions including anthropogenic, biogenic, and shipping emissions. MEIC and REAS v2.2 include only anthropogenic emissions. REAS v2.1 includes all emissions except for lightning by adding the aviation and international shipping emissions from EDGAR v4.2. These differences make the comparison inconsistent. To conduct consistent comparisons, we divide the domain into urban and rural areas based on land-use information and compare the emissions over each area. We use the land-use information at a resolution of 300 m from the GlobCover Land Cover (GCLC version 2.3) database, which is updated for the year 2009. We assume that an urban grid cell has more than 5% coverage of urban land. In GCLC, urban areas are



defined as artificial surface and associated areas. If the grid cell is 100% covered by vegetation (including agricultural land, grass land, shrubs, forest) or barren land, it is regarded as a rural grid cell. We note that the anthropogenic emissions can still be the dominant source over the rural grid cells because urban areas (for example highway, small factories etc.) can be smaller than the resolution of the GCLC data (300 m). Figure 8 shows the distribution of the urban and rural grid cells. The urban grid cells account for 4.5% of the total land area while the rural grid cells cover about 41%.

Figure 9 shows that the meridional distributions of emissions from all inventories over urban grid cells are in good agreement. Even though the urban grid cells are only about 4.5% of total land cover, they contribute to about 50% of the total emissions over the whole domain. The range of the total urban emissions is between about 2.0 (REAS v2.2) to 3.4 (MIX) Gg N year$^{-1}$. The standard deviation of the total urban emissions is about 19% of the ensemble mean. The urban emissions from DECSO-OMI is apparently lower than others above 40 N°. The urban emissions over the areas above 40 N° contribute to only 6% to the total urban emissions over the whole domain in DECSO-OMI; the other inventories show larger contributions (about 10-15%).

Figure 10 shows that the emissions of all inventories over rural grid cells have wide spread along latitudes. The emissions over rural areas contribute to about 8% of the total emissions (254 - 900 Gg N year$^{-1}$) on average with a relative standard deviation of about 44% to the total emissions over the domain. EDGAR v4.3.1 emissions, which include biogenic (biofuel and biomass) emissions, are even lower than MIX and REAS v2.2 that include only anthropogenic emissions. The relative differences in emissions over rural areas are much larger than over urban areas, especially for the bottom-up inventories. These results suggest that large uncertainties over rural areas exist in the current bottom-up inventories. The mean differences are usually smaller for the satellite-derived inventories.

Table 3 and 4 show the temporal correlation of monthly emissions among different inventories over urban and rural areas, respectively. The correlation coefficients are similar between the total emissions over land (Table 2) and over urban areas (Table 3). The two DECSO inventories have a high correlation over rural areas with REAS v2.1. A possible reason is that REAS v2.1 is the only bottom-up inventory in the comparison, which includes biogenic emissions. The EnKF datasets have low correlations with all bottom-up inventories probably due to its coarse horizontal resolution (see Section 3.1.2).

## 4. Discussion

### 4.1 Temporal analyses

Our intercomparison shows differences of about 10% in the time series over of the regional total emissions over Mainland China. The peak year of the Chinese country-total emissions was 2011 in the EnKF and MEIC inventories, which is consistent with the results of Liu et al. (2016). Irie et al. (2016) showed that $NO_2$ column concentrations derived from OMI observations started to decrease in 2011 in China. However, DECSO shows the peak year in 2012, which is consistent with the start of strict air quality regulations (van der A et al., 2017). Over South Korea, the local CAPSS inventory shows




unchanging emissions until 2010 and increasing emissions since. REAS and EDGAR inventories show a strong decrease from 2005 to 2010 with large differences in the decrease rate. For bottom-up inventories, it is difficult to assume the correct timing of penetration of a given new technology. For instance, EDGAR assumes an immediate implementation of a new policy, e.g. low-$NO_x$ burner, and so

does not represent the real time of penetration of the new technology. All the satellite based inventories show a decrease in 2015. This is different for concentrations derived from OMI measurements, which show a decrease from 2005 to 2013 and a slight increase in 2015 (Irie et al., 2016). Mijling et al. (2013) calculated that the $NO_2$ transported from outside the country contributes to about 7% of the average $NO_2$ columns over South Korea. Thus the trend of $NO_2$ column concentrations cannot always be used

as an indication for the local emission because of influences of atmospheric transport. Miyazaki et al. (2017) indicated that an accurate emission trend requires an emission-concentration relationship that explicitly accounts for tropospheric chemistry and transport, which is included in advanced data assimilation techniques. This can cause significant differences between the bottom-up and satellite-derived emissions.

Three satellite-derived inventories (EnKF-CHASER, EnKF-MIROC and DECSO-OMI) and REAS v2.1 show a strong summer peak, which can be attributed to biogenic emissions in summer, such as enhanced soil emissions and emissions from biomass burning. REAS v2.1 includes biogenic emissions while the other bottom-up inventories shown in Figure 6 include only anthropogenic emissions. Zhao and Wang (2009) concluded that soil emissions are enhanced in summer and contribute to at least 14%

of the total emissions in July in China. Soil emissions have a positive exponential relationship with soil temperature (Schindlbacher et al., 2004), and can be enhanced after precipitation over vegetation (Zörner et al., 2016), which could explain the seasonality in satellite-derived emissions in most parts of China. Also, emissions by biomass burning and crop burning are maximized in summer in China (Li et al., 2016; Zhang et al., 2016). Stavrakou et al. (2016) pointed out that the current biomass burning

inventories are underestimating emissions by crop burning in northeast China. This can lead to significant differences in the emission seasonality between the bottom-up and satellite-derived emissions. The OMI satellite instrument with an overpass time in the afternoon will observe more $NO_2$ from biomass burning than the GOME-2 instrument with its morning overpass time, because of stronger biomass burning activities and high soil emissions in the afternoon (Boersma et al., 2008).

This can explain why only the satellite-derived emission inventories based on OMI show a clear summer peak. The seasonality shown by DECSO-OMI and EnKF datasets are slightly different. One possible reason is the uncertainty in the $NO_x$ chemical lifetime in the CTM. For instance, in summer, a too short lifetime can result in overestimation of emissions by the inversion method (Stavrakou et al., 2013). In addition, the EnKF data assimilation has frequent updates (every 120 minutes), which may

lead to overcorrection in the estimated emissions due to the persistent underestimation of $NO_x$ concentrations by the CTM.

### 4.2 Spatial analyses

All the inventories are in good agreement on the spatial distribution on a provincial level but with large

biases. The differences in total annual emissions can increase to 100% for small spatial scales, which is





in line with recent studies on regional emission inventories. For example, Saikawa et al. (2016) compared five bottom-up inventories on a provincial and national scale in China and concluded that more improvements are needed for provincial emissions. All satellite-derived emissions are lower than that of bottom-up inventories at high latitudes (above 40 N°). This is probably due to negative bias of

satellite observations at higher latitudes in East Asia (Ding et al., 2017; Lorente et al., 2017). Another reason can be the overestimation of $NO_x$ lifetime in the CTM for high latitudes (Stavrakou et al., 2013), which could cause underestimation of the estimated emissions.

### 4.3 Differences in the bottom-up emissions

The differences in emission activity and emission factors of each emission sector can lead to large

discrepancies in the bottom-up emission inventories. Saikawa et al. (2016) and Li et al. (2017) analysed that large discrepancies in bottom-up emission estimates over China are mainly related to different statistical data and emission factors for the energy and transport sectors. All three inventories use different provincial statistics. The emission factors of power plants are lower in MEIC as compared to EDGAR and REAS. More power plants are included in MEIC, which uses the high-resolution database

CPED for China. REAS uses two other global databases, CARMA (Carbon Monitor for Action) and WEPP (World Electric Power Plants), while EDGAR uses downscaled IEA (International Energy Agency) statistics and CARMA v3.1 for the distribution. For the transport sector, each inventory has its own sub-categories to calculate the total emissions. MEIC includes only on-road vehicle emissions, while EDGAR and REAS give off-road emissions including emissions from ships and aircrafts. For

vehicle emissions, MEIC uses the information from local statistics, but large uncertainties still exist in vehicle emission factors and activities. The selected vehicle emissions factors do not include spatial variability and the emissions activities are based on surveys conducted in only a few cities (Zheng et al., 2014). More details about the difference in bottom-up inventories can be found in Li et al. (2017). For South Korea, the $NO_x$ emissions from CAPSS, using local data of emission factors and activities based

on local measurements, are much lower than from other bottom-up inventories.

### 4.4 Differences in the satellite-derived emissions

The differences in the satellite-derived emissions are introduced by differences in inversion techniques, chemical transport models, and observations. For instance, errors in the simulated $NO_2$ columns are largely influenced by the representation of lightning $NO_x$ sources, especially in summer, and thus affect

the quality of surface $NO_x$ emissions estimates (Lin, 2012). In the EnKF analyses, lightning $NO_x$ sources are simultaneously optimized using satellite measurements (Miyazaki et al., 2014), which can improve the surface emission estimation. In the DECSO algorithm, the $NO_2$ concentrations above 500 hPa were obtained from climatological data, and the lightning emissions are identified as surface emissions. The error in $NO_x$ lifetime due to uncertainties in meteorological input and chemical

processes also strongly affect the satellite-derived emission estimates (Lin et al., 2012; Stavrakou et al., 2013), which are differently represented in each model.

The emissions derived with DECSO using OMI are much lower than those from other satellite-derived inventories. The monthly variability is largely affected by changing the satellite observations in DECSO, which is attributed to the differences in the satellite overpass time (see section 4.1) and



uncertainty in modelled diurnal variations. $NO_2$ columns observed by GOME2 are on average 20% higher than by OMI in the winter period (October to March) while up to 40% in the summer period (April to September) in 2014, reflecting stronger daytime photochemical sinks from oxidation by OH in summer (Boersma et al., 2009). Although the OMI and GOME-2 retrievals were produced using the same retrieval algorithm (Boersma et al., 2011), Wang et al. (2016) concluded that GOME2-A/B has a larger bias (about 30%) than OMI satellite observations in Wuxi in China from 2011 to 2014. In addition, GOME-2 observes larger $NO_2$ columns from the transport sector due to the morning rush hour (Wang et al., 2016), whereas OMI has better capability to detect emissions from biomass burning that usually at their maximum in the afternoon (Boersma et al., 2008; Miyazaki et al., 2017). To obtain better $NO_x$ emission estimates from the satellite-based approach, it is useful to combine observations from multiple instruments obtained at different overpass time, as implemented in the EnKF inversions. Miyazaki et al. (2017) demonstrated that the application of a correction scheme for diurnal emission variability using multiple measurements (OMI, GOME-2, SCIAMACHY) is an important development in the emission estimates. Different from the DECSO algorithm, the EnKF emissions are obtained by assimilating multiple species, which provide constraints on various aspects of the tropospheric chemistry system to improve the $NO_x$ emission estimates (Miyazaki and Eskes, 2013; Miyazaki et al., 2017). When comparing two $NO_x$ emissions of EnKF-MIROC derived by assimilating multiple species and $NO_2$ only, the average bias is about 5% for the annual total emissions over East Asia. This confirms that the emissions are strongly constrained by $NO_2$ observations but are also modified by non-$NO_2$ observations through their influences on the vertical profile and chemical lifetime of $NO_x$.

EnKF-CHASER and EnKF-MIROC use the same data assimilation framework but different CTMs. Our comparisons reveal that the estimated emissions are largely sensitive to the forecast model. The bias between the two EnKF datasets is about 15% for the annual total emissions over the domain. The estimated emissions are generally higher in EnKF-CHASER than in EnKF-MIROC, which could be attributed to the larger amount of OH and thus shorter $NO_x$ chemical lifetime in EnKF-CHASER.

DECSO inventories show better temporal correlation with other inventories, especially over rural areas. Our analyses suggest that the estimated emissions are strongly influenced by the choice of model resolution (0.25° in DECSO vs. 2.8° in EnKF). Insufficient model resolution could cause artificial dilution and errors in simulating non-linear chemical feedback of $O_3$-$HO_x$-$NO_x$ chemical system. Valin et al. (2011) also discussed that a sufficient horizontal resolution of model to accurately simulate the non-linear effect is related to the size of the emission source (4km for small source to 48km for a big city). High-resolution inversion is considered to be essential to improve the emission estimates at small-scales and to derive better temporal changes over different types of areas.

## 5. Conclusions

To investigate the uncertainties in $NO_x$ emission estimates over East Asia, we compared nine NOx emission inventories from satellite-derived and bottom-up approaches over East Asia. The bottom-up inventories are obtained from a global inventory (EDGAR v4.3.1), regional inventories over Asia (REAS v2.1 and an interim version of 2.2), and national inventories of China (MEIC) and South Korea (CAPSS). The four satellite-derived inventories were derived with different versions of DECSO and





EnKF. DECSO-OMI and DECSO-GOME2a are based on an extended Kalman filter using observations from OMI and GOME-2 respectively. EnKF-CHASER and EnKF-MIROC are based on an ensemble Kalman filter applied to multiple-$NO_2$ and multiple-species measurements using two different CTMs. The intercomparison of all inventories shows good agreement in $NO_x$ emissions over Mainland China

with an average bias of about 20% for yearly emissions. All the inventories reveal common variability such as a typical emission reduction of 10% during the Chinese New Year and a sharp peak in December. All the inventories show good agreement in spatial distribution on a provincial level (R>0.8). However, biases in $NO_x$ emissions and uncertainties in temporal (both yearly and monthly) variability become quickly higher when the spatial scale decreases. The emissions over urban areas in

East Asia and show better agreement in the temporal and spatial variability between the different inventories than over rural areas. The coarse-resolution EnKF inventories show low temporal correlation with other inventories over rural areas, because they are unable to distinguish between urban and rural sources at small-scale. All the satellite-derived inventories except for DECSO-GOME2a show a summer peak over Mainland China. The summer peak could be related to enhanced

emissions from soils and biomass burning in summer, for which OMI better captures the enhanced concentrations than GOME-2 due to their different overpass time. All the satellite-derived inventories, in particular DECSO-OMI, are lower than bottom-up inventories at higher latitudes (at above 40 N°).

Based on our findings from the intercomparison of $NO_x$ emission inventories, we come to the following recommendations for the development of $NO_x$ emission inventories in the future:

1. To better capture the temporal variability of emissions from the satellite-derived approach, observations from multiple instruments are important. In addition, a high spatial resolution model is necessary to distinguish between different emission sources and to better derive the emissions on a small spatial scale.

2. Different bottom-up inventories use different definition of emission sector categories and with
different assumption about the shares of the different technologies and high uncertainties in $NO_x$ emission factors and activities. Better measurements of emissions factors and more detailed statistics on the activities not only per fuel but also per (combustion) technology are required.

3. To take into account the advantages of bottom-up and satellite-derived approaches and to
update near-real time emissions, sector information from the bottom-up approach can be combined with the satellite-derived inventories. For instance, information on temporal changes of emissions from satellite-based calculations can be used for the temporal evolution of the bottom-up inventories

**Acknowledgements**

The research was part of the OMI project funded and supported by the Netherlands Space Office. We acknowledge IPSL/LMD, INERIS and IPSL/LISA in France for providing the CHIMERE model. We acknowledge the use of tropospheric $NO_2$ column data obtained from [www.temis.nl](www.temis.nl) and the ESA GlobCover 2009 Project for the land use data set. We thank Henk Eskes and Folkert Boersma for their helpful suggestions.





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





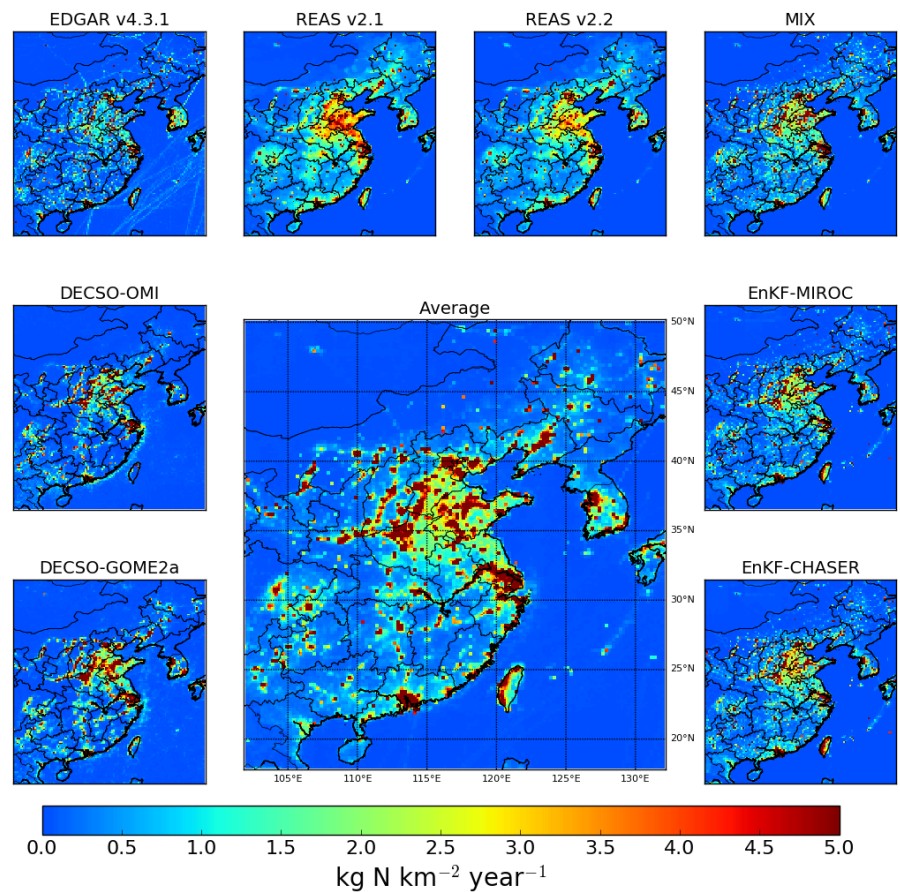

**Figure 1.** NO$_x$ emissions over East Asia of all inventories in this study and their average in 2008. The averaged emissions over land are the mean of all emission inventories. Over the ocean the emissions are the average of DECSO-OMI, DECSO-GOME2, REAS v2.2 and EDGAR v4.3.1.





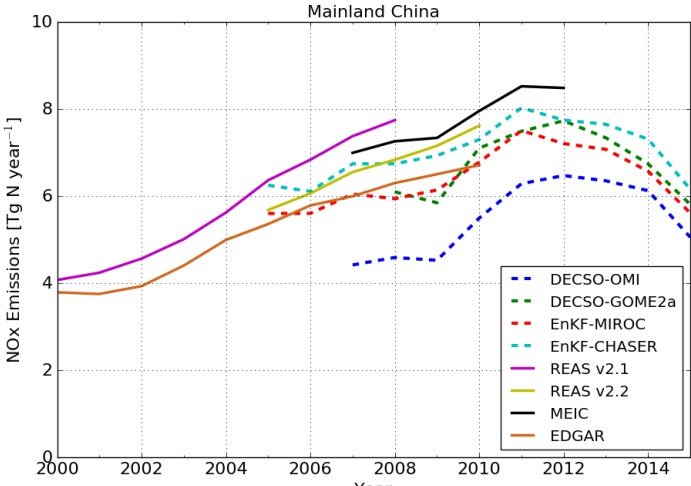

**Figure 2. Annual NO$_x$ emissions from eight emission inventories over Mainland China in the selected domain (Figure 1).**



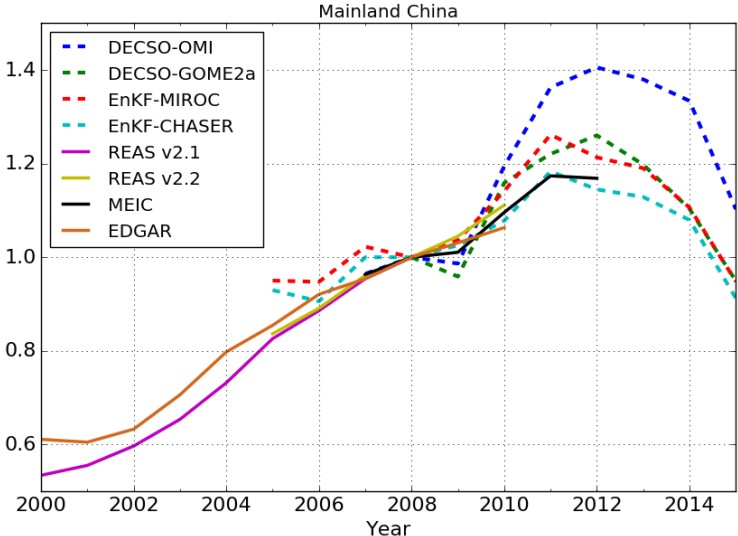

**Figure 3. Normalized annual NO$_x$ emissions from eight emission inventories over Mainland China in the selected domain (see Figure 1). The time series are normalized to their value in 2008.**





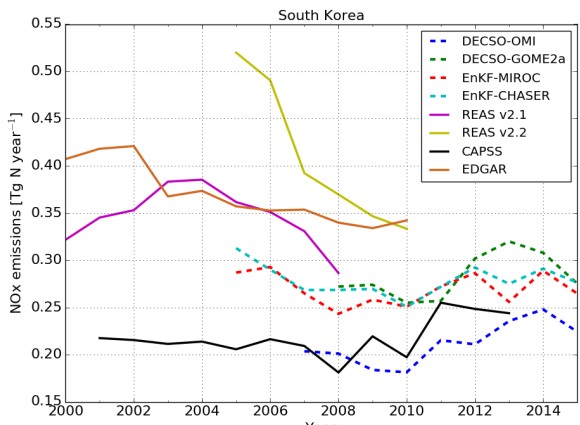

**Figure 4. Annual NOₓ emissions from eight inventories over South Korea.**





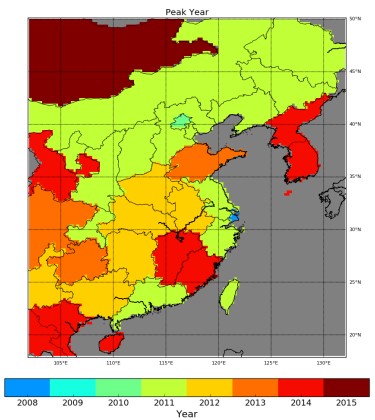 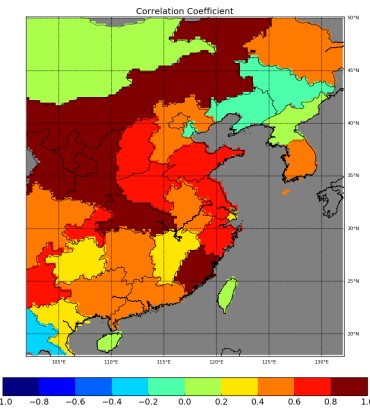

**Figure 5. The peak year (left figure) of the average annual NO$_X$ emissions from the four satellite-derived inventories (DECSO-OMI, DECSO-GOME2a, EnKF-CHASER and EnKF-MIROC) for various provinces and regions. The right map image shows the minimum temporal correlation coefficient of all combinations of the four emission datasets.**





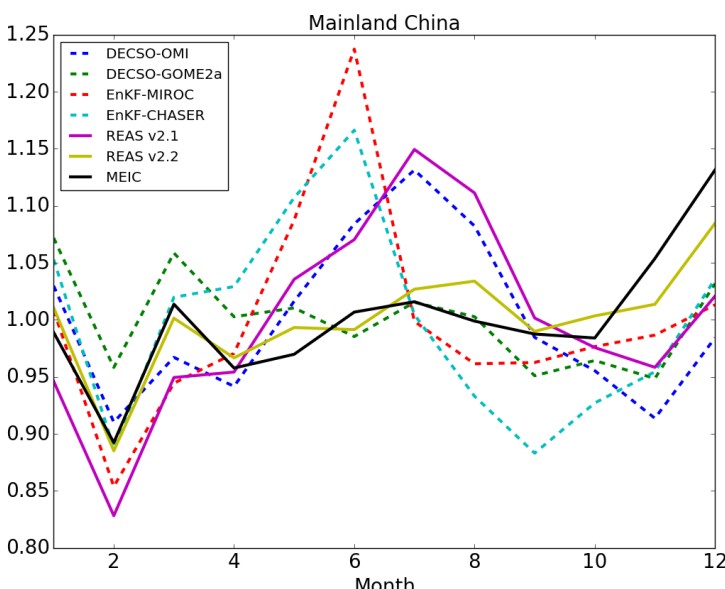

**Figure 6. Monthly variability in NO$_x$ emissions over Mainland China in the selected domain (see Fig 1).**





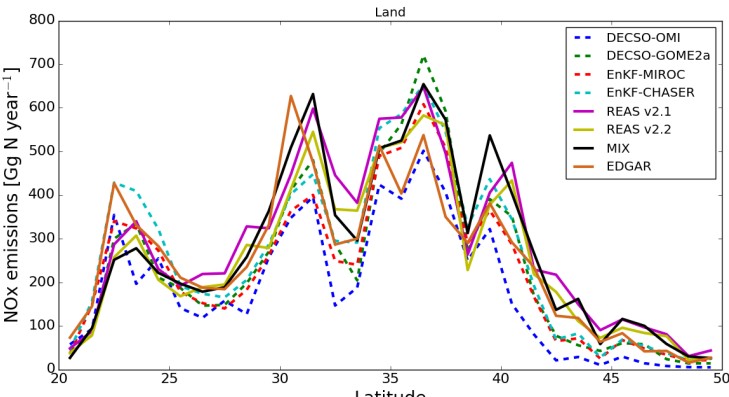

**Figure 7. Latitudinal distribution of longitudinal total NO$_x$ emissions over land. The emissions are summed over a one degree longitudinal band above land.**





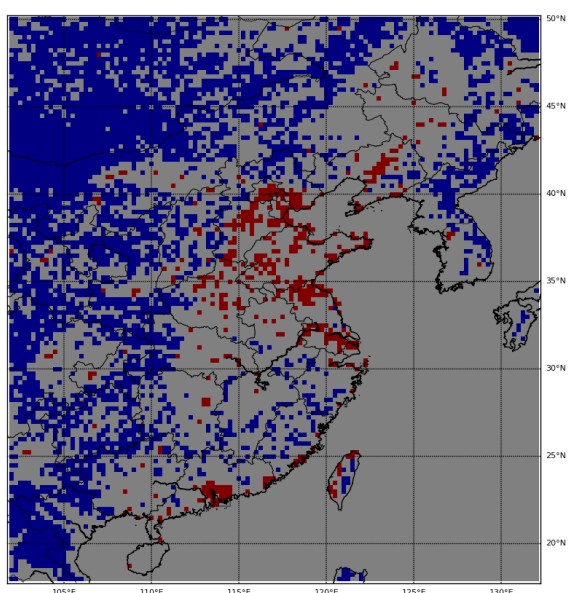

Figure 8. The distribution of urban grid cells (red) and vegetation grid cells (blue). Urban grid cells have an urban area (as defined in the GCLC data base) larger than 5% of the grid cell. Vegetation grid cells are 100% covered by vegetation.





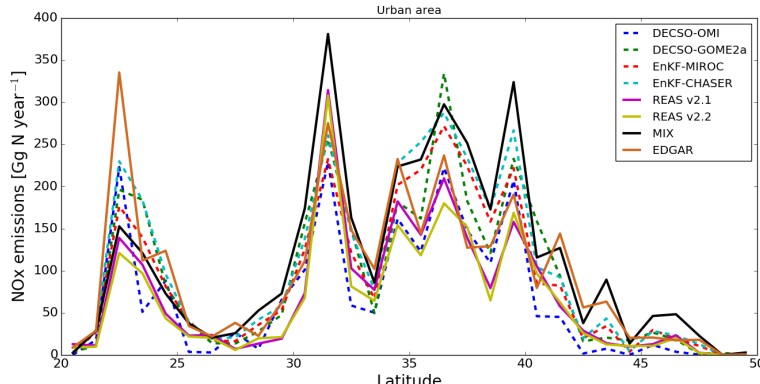

**Figure 9. Latitudinal distribution of longitudinal total NOx emissions over urban areas (the red cells in Fig 8).**





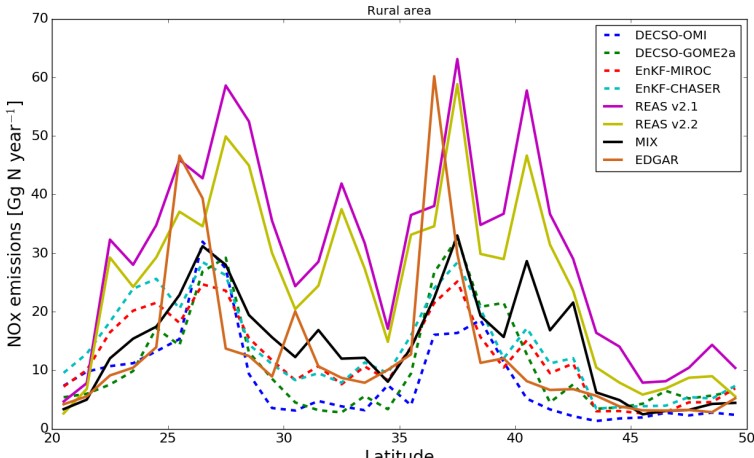

Figure 10. Same as Fig. 9, but for rural grid cells (the blue cells in Figure 8).



Table 1 Temporal correlation coefficient of monthly NO$_x$ emissions over Mainland China from 2008 to 2010. The bold rectangle indicates the relation between bottom-up and satellite-based inventories.

| | REAS v2.2 | MEIC | EnKF-MIROC | EnKF-CHASER | DECSO-GOME2a |
|---|---|---|---|---|---|
| DECSO-OMI | 0.72 | 0.6 | 0.52 | 0.45 | 0.75 |
| DECSO-GOME2a | 0.78 | 0.74 | 0.46 | 0.28 | 1 |
| EnKF-CHASER | 0.43 | 0.43 | 0.65 | 1 | - |
| EnKF-MIROC | 0.46 | 0.43 | 1 | - | - |
| MEIC | 0.91 | 1 | - | - | - |





Table 2. Spatial correlation coefficients of annual NO$_x$ emissions in 2008 over land on a provincial level in china and other regions outside China. (The total number of provinces and regions is 33, which includes including 29 Chinese provinces, South Korea, North Korea, and parts of Mongolia (83%) and Vietnam (43%), see figure 1. The study domain only covers part of the provinces Heilongjiang (74%), Inner Mongolia (83%) and Yunnan (36%)). The bold rectangle indicates the relation between bottom-up and satellite-based inventories.

|  | EDGAR v4.3.1 | REAS v2.1 | REAS v2.2 | MEIC | EnKF-MIROC | EnKF-CHASER | DECSO-GOME2a |
|---|---|---|---|---|---|---|---|
| DECSO-OMI | 0.84 | 0.90 | 0.91 | 0.92 | 0.96 | 0.95 | 0.98 |
| DECSO-GOME2a | 0.85 | 0.92 | 0.94 | 0.95 | 0.96 | 0.96 | 1 |
| EnKF-CHASER | 0.88 | 0.92 | 0.93 | 0.95 | 1 | 1 | - |
| EnKF-MIROC | 0.88 | 0.94 | 0.95 | 0.95 | 1 | - | - |
| MEIC | 0.89 | 0.99 | 0.98 | 1 | - | - | - |
| REAS v2.2 | 0.90 | 0.99 | 1 | - | - | - | - |
| REAS v2.1 | 0.88 | 1 | - | - | - | - | - |
| EDGAR v4.3.1 | 1 | - | - | - | - | - | - |



Table 3 Temporal correlation coefficient of monthly total NO$_x$ emissions over urban areas in the common year 2008. The bold rectangle indicates the relation between bottom-up and satellite-based inventories.

| | REAS v2.1 | REAS v2.2 | MIX | EnKF-MIROC | EnKF-CHASER | DECSO-GOME2a |
|---|---|---|---|---|---|---|
| DECSO-OMI | 0.73 | 0.61 | 0.34 | 0.5 | 0.22 | 0.69 |
| DECSO-GOME2a | 0.53 | 0.67 | 0.48 | 0.72 | -0.04 | - |
| EnKF-CHASER | 0.28 | 0.21 | 0.47 | 0.21 | - | - |
| EnKF-MIROC | 0.11 | 0.24 | 0.25 | - | - | - |
| MIX | 0.72 | 0.83 | - | - | - | - |
| REAS v2.2 | 0.93 | - | - | - | - | - |





Table 4 Same as Table 3, but for rural areas in 2008.

| | REAS v2.1 | REAS v2.2 | MIX | EnKF-MIROC | EnKF-CHASER | DECSO-GOME2a |
|---|---|---|---|---|---|---|
| DECSO-OMI | 0.88 | 0.17 | -0.31 | 0.75 | 0.64 | 0.69 |
| DECSO-GOME2a | 0.88 | 0.33 | -0.21 | 0.12 | 0.20 | - |
| EnKF-CHASER | 0.46 | -0.02 | -0.12 | 0.75 | - | - |
| EnKF-MIROC | 0.47 | 0.20 | 0.04 | - | - | - |
| MIX | -0.13 | 0.80 | - | - | - | - |
| REAS v2.2 | 0.43 | - | - | - | - | - |