# Peer review of "Intercomparison of $NO_x$ emission inventories over East Asia"

_Atmospheric Chemistry and Physics, 2017_

## Referee Comment (RC1) · Anonymous Referee #2 · 30 May 2017

The paper compares NOx emission inventories that use bottom-up methods as well as satellite remote sensing. This shows the relative merits of the methods and underscores the need for using multiple satellite sensors in developing inventories. The paper is clear and thorough and within the scope of ACP, I am happy to recommend publication subject to minor comments listed below.

Page 11, Line 25: Temporal correlation coefficients in Tables 3 & 4 are similar to the spatial coefficients in Table 2? Do you mean the pattern is similar? In general, I think it would be good to add the number of points used in the correlation coefficients for the different tables. This would help interpret the r values and also clarify what they are based on.

Table 3 & 4: If I understood correctly, these are based on 12 data points similar to those

in Fig. 6? I think Fig. 6 could be expanded to show total emissions for Urban areas (panel b) and Rural areas (panel c), keeping panel a as it is.

The three bullet points in the conclusions seem a little disconnected from the body of the paper. It may be useful to add some discussion of how the present study relates to the expected products of the GEMS sensor on GEO-KOMPSAT-2B.

Technical comments: Fig 5: The labels are not legible (remove lat/lon, increase font size for title and colorbar, label colorbar on right).

Table 2 caption: China should be capitalized.

---

## Referee Comment (RC2) · Anonymous Referee #1 · 9 Jun 2017

General comments This paper presents a comparison of nine NOx emission inventories for East Asia including five bottom-up and four top-down inventories constrained by satellite observations, shows large uncertainties associated with their spatial and temporal variabilities, and suggests a few important issues for future development of an emission inventory in this region. The paper is generally well written but can be improved significantly after a careful editing.

I believe that this kind of intercomparison study for emission inventories in East Asia has not extensively been conducted yet. In this regard, this manuscript includes interesting and important results, which are worthy of publication. However, I have a major concern in the analysis used in this paper, which prevents me from recommending acceptance with ACP at the present form, and I will elaborate it below.

[Figure]

As indicated in the manuscript, some inventories include both anthropogenic and natural sources, but some do not have natural sources such as soil, lightning, and biomass burning, which have large seasonal and inter-annual variabilities. Without carefully segregating natural source contributions to the inventories, a simple comparison of total values among the inventories may cause a serious misunderstanding in their quality. I believe that all bottom-up inventories have sectoral emissions. The top-down emissions with CTMs may have a difficulty to separate individual sources but at least could separate anthropogenic versus natural sources because they have different seasonal variation. Therefore, I suggest that authors should compare inventories including anthropogenic sources alone and then go on to do the similar analysis for natural sources separately if they can.

Specific comments 1) Page 1, line 30, - I have a hard time to agree with this sentence because it does not appear that they show good agreement in total values.

2) Page 2, line 1, - There must be some typos here.

3) Section 2 – I believe that each inventory typically has a base year from which it projects values for other years based on some proxy data. If there is available information on this, please state it in the manuscript.

4) Page 6, line 15 – CHIMERE has a top layer at 500 hPa, which is too low to account for lightning NOx emission. So in the inverse modeling with CHIMERE, how would lightning NOx contribution to the observations be taken into account? I would assume that a climatological partial column would not change with time.

5) Page 6, line 28 – Obviously, there is a difference in the pixel sizes of satellite observations, which also differ from the model resolution. A detailed information on this would be necessary in the manuscript. How would this difference cause a discrepancy in the DECSO data with each satellite observation?

6) Page 7, line 35 – revision is required for clarity.

7) Page 8, line 10 – Could you explain the reason for the discrepancy here?

8) Page 9, line 30 – How would the difference in the local overpass time and pixel size make a difference in the seasonality of inferred emissions? It is not clear to me at all.

9) Page 10, line 29 – "Biases" would not be appropriate because we don't know the true. I would suggest to use "differences" instead.

10) Page 10, line 32 – How about lightning?

11) Page 11, line 7 – It appears to me the same as in Figure 7.

12) Page 14, line 21 – It makes me wonder if two simulated NO2 concentrations also show a similar or greater magnitude of differences as shown in the top-down emissions.

---

## Author Comment (AC2) · 4 Jul 2017

We appreciate reviewer #1 for giving valuable comments. We respond to each specific comment below and indicate what changes we have made in the manuscript. The comments and questions from the referee are in blue and italic font.

*As indicated in the manuscript, some inventories include both anthropogenic and natural sources, but some do not have natural sources such as soil, lightning, and biomass burning, which have large seasonal and inter-annual variabilities. Without carefully segregating natural source contributions to the inventories, a simple comparison of total values among the inventories may cause a serious misunderstanding in their quality.*
*I believe that all bottom-up inventories have sectoral emissions. The top-down emissions with CTMs may have a difficulty to separate individual sources but at least could separate anthropogenic versus natural sources because they have different seasonal variation. Therefore, I suggest that authors should compare inventories including anthropogenic sources alone and then go on to do the similar analysis for natural sources separately if they can.*

Thank you for the suggestion. Based on EDGAR and REAS v2.1, we calculate the ratio of biogenic emissions and the anthropogenic for the study domain. The ratio is about 4.5% and 6.5%, which means that the anthropogenic is the dominant source for the study domain. Separation of the anthropogenic and natural source for the satellite-derived emissions will introduce large uncertainties (larger than the ratio) resulting in an unfair intercomparison at present. On page 10 line 32, we analyze the inconsistency of the intercomparison. For this reason, we have separately analyzed the emissions over urban and rural areas. In section 4.3, we have discussed the different definitions of source categories in the bottom-up inventories.
We add the following discussion in section 3.2 on page 11 line 2:
"The seasonal cycle of the urban grid cells is quite similar to the one shown in Figure 6 for all of mainland China. However, the seasonal cycle for the rural grid cells have a much stronger summer maximum for the inventories that include biogenic emissions"

*Specific comments 1) Page 1, line 30, - I have a hard time to agree with this sentence because it does not appear that they show good agreement in total values.*

We conclude that the total values are in good agreement because the uncertainty of NOx emissions of each inventory is also quite large in the current stage. For example, Li et al. (2017) showed that the uncertainty of NOx is about ±31% in MIX for China (MEIC). The uncertainty of NOx in REAS v2.1 is about ±37% over China (Kurowaka et al. 2013). The standard deviation of total emissions is about 20% for the common year 2008. The distribution is in the range of the uncertainties of these two frequently-used inventories (MIX and REAS v2.1) by modelers for this region. Thus, we conclude that the total values are in good agreement.
We add this discussion on Page 8 line17:
"For comparison, Li et al. (2017) reported typical uncertainties of 31 to 37 % for bottom-up inventories over China."

*2) Page 2, line 1, - There must be some typos here.*
We noticed that the first part of the sentence was missing on Page 4.
We add the missing part in the paper:
"In this study, we compare the satellite-derived inventories from DECSO and EnKF with a…"

*3) Section 2 – I believe that each inventory typically has a base year from which it projects values for other years based on some proxy data. If there is available information on this, please state it in the manuscript.*

According to our information, all bottom-up inventories in our paper use full time series of the emission activity data. They all calculate the emissions from year to year and they don't use projection. However, the spatial proxy data that it uses for the geospatial distribution does not vary a lot in time. To avoid confusion, we prefer not to mention this detailed information in our paper.

*4) Page 6, line 15 – CHIMERE has a top layer at 500 hPa, which is too low to account for lightning NOx emission. So in the inverse modeling with CHIMERE, how would lightning NOx contribution to the observations be taken into account? I would assume that a climatological partial column would not change with time.*

The DECSO algorithm underestimates lightning emissions for those reasons. The algorithm is not able to capture the emissions with temporary changes less than one day, such as lightning emissions. This have been discussed in the paper of Ding et al. (2017). In the manuscript (page 12 line 32), we explained:

"In DECSO algorithm, the $NO_2$ concentrations above 500hpa were obtained from climatological data, and the lightning emissions are identified as surface emissions. "

For clarification, we add the following text on page 6 line 20:

"NOx emissions detected with DECSO are regarded as total surface emissions. Note that the algorithm is not able to capture the emission with temporary changes less than one day."

*5) Page 6, line 28 – Obviously, there is a difference in the pixel sizes of satellite observations, which also differ from the model resolution. A detailed information on this would be necessary in the manuscript. How would this difference cause a discrepancy in the DECSO data with each satellite observation?*

We project the model grid cells on the satellite footprint in DECSO. For large areas, the effect of different footprint size is averaged out. The different total emissions derived from the two satellite instruments are mainly caused by the different overpass time of satellites and the diurnal cycle in the model (see point 8). The effect of the pixel sizes of satellite observations is mainly for small areas with inhomogeneous land-use. The resolution of emissions cannot be higher than the satellite pixel size.

*6) Page 7, line 35 – revision is required for clarity.*

We change the sentence to:

"For the comparison, the monthly emissions on 2.8° resolution from EnKF-CHASER and EnKF-MIROC are redistributed to a horizontal resolution of $0.25° \times 0.25°$ based on emission distributions of the a priori inventory MIX. Note that the shipping emissions near the coast areas are added to land since they are not included in MIX."

*7) Page 8, line 10 – Could you explain the reason for the discrepancy here?*

The possible reasons are related to the uncertainties in model diurnal cycles and satellite observations, which are discussed in section 4.4.

*8) Page 9, line 30 – How would the difference in the local overpass time and pixel size make a difference in the seasonality of inferred emissions? It is not clear to me at all.*

We have explained this in the discussion part.

"The OMI satellite instrument with an overpass time in the afternoon will observe more NO2 from biomass burning than the GOME-2 instrument with its morning overpass time, because of stronger biomass burning activities and high soil emissions in the afternoon (Boersma et al., 2008). This can explain why only the satellite-derived emission inventories based on OMI show a clear summer peak."

*9) Page 10, line 29 – "Biases" would not be appropriate because we don't know the true. I would suggest to use "differences" instead.*
We change it to 'difference'

*10) Page 10, line 32 – How about lightning?*
We have neglected lightning emissions in the whole analysis. The datasets from EnKF are only surface emissions. The lightning part of EnKF is not used in this study. The DECSO algorithm underestimates lightning emissions. Lightning emissions are shorter time scale and are averaged out in the monthly emissions.

*11) Page 11, line 7 – It appears to me the same as in Figure 7.*
Figure 9 shows the emissions over the urban area, which is different from the total emissions shown in Figure 7. Since the emissions over the urban area are dominant (about 50% of the total emissions), these look similar. However, the influences of rural emissions are not negligible, and it is worth showing Figure 9.

*12) Page 14, line 21 – It makes me wonder if two simulated NO2 concentrations also show a similar or greater magnitude of differences as shown in the top-down emissions.*
The difference of NO2 concentrations simulated by CHASER and MIROC is about 10% over this region.

---

## Author Comment (AC1)

We appreciate reviewer #2 for giving valuable comments. We respond to each specific comments below and indicate what changes we have made in the manuscript. The comments and questions from the referee are in blue and italic font.

*Page 11, Line 25: Temporal correlation coefficients in Tables 3 & 4 are similar to the spatial coefficients in Table 2? Do you mean the pattern is similar? In general, I think it would be good to add the number of points used in the correlation coefficients for the different tables. This would help interpret the r values and also clarify what they are based on.*

Thank you for pointing out the comparison. It is a typo here. We intend to show the temporal correlation coefficients in table 3 over urban area are comparable with the temporal correlation coefficients in table 1. The sentence should be that "The correlation coefficients are similar between the total emissions over land (Table 1) and over urban areas (Table 3)."
In table 3 and 4, for the temporal correlation coefficient, the calculations are based on the monthly emissions of the year 2008, which are only 12 numbers. In table 1, we use monthly emissions from 2008 to 2010, therefore, the correlation coefficients are calculated based on 36 numbers. The analysis of Table 3 and 4 is an example to show that the uncertainties are larger over rural areas, not only on the spatial distribution but also on the temporal changes. We included the number of points that the R-values are based on in the table caption.

*Table 3 & 4: If I understood correctly, these are based on 12 data points similar to those in Fig. 6? I think Fig. 6 could be expanded to show total emissions for Urban areas*
*(panel b) and Rural areas (panel c), keeping panel a as it is.*

The seasonality showed in Figure 6 is averaged based on the time period of each inventories. This has been described on Page 9 Line 11. The temporal correlation coefficients in Table 3 and 4 calculated for urban and rural areas are based on year 2008. This analysis is an example to show that the uncertainties are larger over rural areas, not only on the spatial distribution but also on the temporal changes. The distinction between rural and urban grid cells is much later introduced in the paper than Figure 6. We would like to keep this sequence and add this discussion in section 3.2 on page 11 line 2:
" The seasonal cycle of the urban grid cells are quite similar to the one shown in Figure 6 for all of mainland China. However, the seasonal cycle for the rural grid cells have a much stronger summer maximum for the inventories that include biogenic emissions."

*The three bullet points in the conclusions seem a little disconnected from the body of the paper. It may be useful to add some discussion of how the present study relates to the expected products of the GEMS sensor on GEO-KOMPSAT-2B.*

We have added some discussion about how the future missions of satellite observations affect the present study at the end of the paper.
"The satellite-derived approach can be further improved following the development of satellite instruments, like TROPOMI on Sentinel 5p and later the GEMS sensor on the geostationary GEO-KOMPSAT-2B. With higher spatial resolution of observations, more accurate emission over different land use categories can be obtained. GEMS will provide observations with high temporal and spatial resolution, which enables the improvement of diurnal cycles for emission estimates."

*Technical comments: Fig 5: The labels are not legible (remove lat/lon, increase font size for title and colorbar, label colorbar on right).*

We have changed it based on the suggestion.

*Table 2 caption: China should be capitalized.*

We have changed it.

*Table 2 caption: China should be capitalized.*